# The Norwegian version of the Copenhagen Psychosocial Questionnaire (COPSOQ III): Initial validation study using a national sample of registered nurses

Solveig Osborg Ose[1]*, Signe Lohmann-Lafrenz[2], Vilde Hoff Bernstrøm[3], Hanne Berthelsen[4], Gunn Hege Marchand[5]

1 SINTEF, Health Services Research group, Trondheim, Norway, 2 NTNU Faculty of Medicine, Department of Public Health and Nursing, Trondheim, Norway, 3 Oslo Metropolitan University, Centre for Welfare and Labour Research, Oslo, Norway, 4 Malmö University, Centre for WorkLife and Evaluation Studies, Malmö, Sweden, 5 NTNU Faculty of Medicine, Department of Neuromedicine and Movement Science, Trondheim, Norway

* solveig.ose@sintef.no

**Data Availability Statement:** All relevant data are within the paper and its Supporting Information files.

## Abstract

### Background

Employers are legally obligated to ensure the safety and health of employees, including the organizational and psychosocial working environment. The Copenhagen Psychosocial Questionnaire (COPSOQ III) covers multiple dimensions of the work environment. COPSOQ III has three parts: a) work environment b) conflicts and offensive behaviours and c) health and welfare. We translated all three parts into Norwegian and evaluated the statistical properties of the 28 work environment dimensions in part a), using a sample of registered nurses.

### Methods

The original English version was translated into Norwegian and back translated into English; the two versions were compared, and adjustments made. In total, 86 of 99 items from the translated version were included in a survey to which 8804 registered nurses responded. Item response theory models designed for ordinal manifest variables were used to evaluate construct validity and identify potential redundant items. A standard confirmatory factor analysis was performed to verify the latent dimensionality established in the original version, and a more exploratory factor analysis without restrictions is included to determine dependency between items and to identify separable dimensions.

### Results

The measure of sampling adequacy shows that the data are well suited for factor analyses. The latent dimensionality in the original version is confirmed in the Norwegian translated version and the scale reliability is high for all dimensions except 'Demands for Hiding Emotions'. In this homogenous sample, eight of the 28 dimensions are found not to be separate

**Funding:** Funding for the data collection was provided by the Norwegian Nurses Organisation (NSF). No specific funding was received for this publication.

**Competing interests:** The authors have declared that no competing interests exist.

dimensions as items covering these dimensions loaded onto the same factor. Moreover, little information is provided at the low and high ends of exposure for some dimensions in this sample. Of the 86 items included, 14 are found to be potential candidates for removal to obtain a shorter Norwegian version.

## Conclusion

The established Norwegian translation of COPSOQ III can be used in further research about working environment factors and health and wellbeing in Norway. The extended use of the instrument internationally enables comparative studies, which can increase the knowledge and understanding of similarities and differences between labour markets in different countries. This first validation study shows that the Norwegian version has strong statistical properties like the original, and can be used to assess work environment factors, including relational and emotional risk factors and resources available at the workplace.

## Introduction

### The importance of labour

Labour is the fundamental resource in a country's economy, and human capital is considered the most prominent input factor in the production of goods and services [1,2]. Labour generates wages and salaries, which finance consumer spending and secure economic performance in a country, while taxation of income from work contributes to covering welfare expenditures [3]. However, management of human capital requires very different systems and routines than other input factors such as land and physical capital. Human capital needs active investment rather than passive management or administration, and the quality of the work environment can be viewed as a key performance indicator for the human capital at the workplace [4]. Investing in the work environment may be profitable because employers need to pay higher wages to compensate for poor working conditions [5], and because improved work environment will potentially increase personnel productivity [6] and reduce sickness absence rates [7].

### Productivity of the labour force and Occupational Health and Safety

Lost productivity due to health problems of employees that could be prevented at the workplace represents an important productivity loss to the individual employee, the employer and to society through poorer economic performance [8–10]. Prevention at the workplace should obviously target work accidents and injuries, but also working conditions that pose long-term health hazards to employees as health deteriorates under poor working conditions [11,12]. The measures that are required to prevent different work-related risks must be assessed systematically at the individual workplace, and these activities represent the Occupational Health and Safety (OHS) practice at each workplace [13]. Planning and conducting OHS activities are essential components of the management of and investment in the human capital at the workplace [14–16].

Most industrial countries have extensive and comprehensive systems of OHS management, and this is demonstrated by consistently reduced work-related accident rates [17,18] and increased productivity [19,20]. It is established that productivity is a key element of the economic attractiveness of investing in OHS [21,22].

## Working life in different countries

Labour markets in different countries impose various requirements on employees, employers, and on national, regional, and local authorities. Access to raw materials and production resources in general, distance to trade markets and other competitive conditions determine which jobs exist in a country. The mixture of jobs stipulates the workforce composition, while historical and current labour relations determine co-operative policy between employers and employees, workers' rights and privileges, the structure of remuneration, and the legal requirements imposed on the working environment and working conditions in general. For instance, countries with strict employment protection and co-ordinated labour relation regimes are found to specialize in industries with a cumulative knowledge base, that is, continuous development of the human capital [23–25]. In contrast, China for instance has dominated the clothing industry because of the low wages and its ability to also keep non-labour costs very low [26].

## Comparative research and validation of instruments across cultures

It is argued that job quality and thus work environment should be a political priority and that high-quality comparative data covering various working environment issues are important to human progress [27,28]. Comparing work environment factors and identifying mechanisms in the labour market across countries can motivate improvements and thereby reduce the adverse health consequences of working. Improved working conditions can imply reduced productivity loss due to sickness absenteeism [29], and increased employment and higher retirement age because more people are able to work, and want to work for more years if the working conditions are improved [30].

However, questionnaires about perceived work environment issues are not necessarily transferable between countries or across times. Comparative research and cross-cultural research in general have specific methodological problems, mostly relating to translation quality and the comparability of results across different cultures and traditions [31]. The use of a previously developed and validated instrument for research can facilitate the building of cross-cultural knowledge [32]. An example of such an instrument is the Copenhagen Psychosocial Questionnaire (COPSOQ), which today is used worldwide in research and for workplace assessments [33].

## Psychosocial and organisational risk factors neglected in many OHS management system

Employers are legally obligated to assess risks in most countries through national OHS legislation. Assessing work environment risk factors can provide indications of the state of OHS practices and identify the need for improvement at the workplace.

The concept of psychosocial work environment has gradually been developed with inspiration from the workplace accident prevention literature that in the 1950s used descriptions as "a social-psychological approach" [34], "psychological work environment" and "psychological climate of the workplace" [35]. In the 1960-70s the industrial democracy research introduced "psychological and social criteria for the evaluation of work situations" and "psychosocial aspects of the work environment" [36,37], while the work stress research at the end of 1970s used the term "psychosocial effects of work environments" [38] and "psychosocial environment" [39].

In the early days, the main emphasis was on "psychosocial hazards", which Coz and Griffiths defined as aspects of the "design and management of work and its social and

organisational contexts that have the potential for causing psychological or physical harm" [40]. Over the years, it has become clear that a strict focus on avoiding risks that may lead to health-related problems is problematic and not necessarily leading to the wanted effects [41]. Thus, Leka et al argue for a need of also addressing psychosocial factors that can promote well-being and performance through organisational learning and development [38]. In line with this, Rugulies more recently suggests that the psychosocial work environment is important for understanding how the "interrelation of societal structures, environmental exposures, and psychological and psycho-physiological processes affect health and illness" [42]. In line with this extended understanding of the research field, the term "psychosocial work environment" has been replaced with "organisational and social work environment" in the newest Swedish regulations; thus, signalling a clear shift towards more emphasis on the contextual work factors rather than individual factors.

Nevertheless, psychosocial risks are among the most challenging risk factors in OHS management [43]. One reason for this is that this type of risk is often not sudden or as acute as the risk factors that traditional OHS management is oriented towards. Psychosocial risks may be acute (e.g., an act of violence), a risk in progress (e.g., bullying), or a risk to health and welfare due to long-term exposure (e.g., continual poor management).

The Copenhagen Psychosocial Questionnaire (COPSOQ) covers a variety of dimensions of the work environment and is an obvious candidate to be established as an important risk assessment tool in Norway. The COPSOQ instrument is available in more than 25 languages, is generic and can be used at any type of workplace. Version III also provides an updated instrument allowing comparability between populations and time periods and the potential for comparative analyses is extensive [33]. Benchmark or population-based reference values to assess the need for preventive action at the workplace are available in several countries [44]. The questionnaire defines mandatory core items that should be included in national short, middle, and long versions of the COPSOQ III instrument [32].

## This study

International collaboration to study exposure to organizational and psychosocial risk factors can contribute to new knowledge and the potential development of effective prevention measures. The collection of data using the same questionnaire across countries and in different labour markets enables comparative analyses, and this has been the motive for translating COPSOQ III into Norwegian and examining the instrument's statistical properties.

The objective of this study was to translate the original English version of COPSOQ III into Norwegian and evaluate the statistical properties of the data collected with the translated version in a large sample of registered nurses. As such, this is the first validation study of COPSOQ III from Norway.

## Method

### COPSOQ III

COPSOQ III has three parts: a) work environment (99 items covering 28 dimensions), b) conflicts and offensive behaviours (16 items covering 7 topics) and c) health and welfare (31 items covering 8 topics) [33] and all 146 items were translated. Part a) is the focus of this study and this part covers multiple dimensions of work environment factors based on well-researched psychosocial theories such as Demand–Control–Support model [38], Effort–Reward Imbalance model [45], Job Demand-Resources model [46] and Workplace Social Capital [47]. The 28 dimensions in part a) are analysed based on 86 out of 99 items.

## Translation

To ensure semantic equivalence, that is, that the meaning of the individual original item is retained in the translated version, the translation was performed using a forward–backward translation process [48,49]. The first translation from English to Norwegian was independently conducted by three native Norwegian researchers (GHM, VHB and SOO) and the process was co-ordinated by the fourth researcher (SLL), a native Danish occupational doctor living and working in Norway. Four workshops were held during a four-week period. The fifth researcher involved in the translation process (HB) is also native Danish and currently working in Sweden. HB has experience from international development of COPSOQ III and from translation and validation of the instrument in Sweden. HB contributed to workshops when different items were discussed to ensure that the intended meaning of the items was retained in the translated version. After consensus was reached on all items, the items were sent to a professional translation bureau that back translated all items to English. The English back-translation was then compared with the original version and revisions were made to the final version. Finally, to avoid potential linguistic problems in the Norwegian version, an experienced professional linguist thoroughly reviewed the questionnaire.

The Norwegian written language has two official norms, *bokmål*, 'book language' and *nynorsk*, New Norwegian, and the items were translated into *bokmål*. There are many spoken dialects in Norway, almost as many as there are places where people live, and the three native Norwegian researchers who independently translated the items all speak different dialects. During the translation process, we discussed some words that were dialect words or dialectical ways of formulating sentences. We avoided all dialect words and we also tried to keep an informal oral style in the translation, for instance, avoiding words that were too formal or technical.

## Data collection and data sample

In September 2021, a national survey among registered nurses was conducted on behalf of the Norwegian Nurses Organisation where all core items, all but one of the middle version items, and 12 of the items in the long version, that is, 133 items were included. A total of 30,070 registered nurses received an invitation to participate in the study and 13,045 (43%) responded. However, 13% (n = 1664) of these did not satisfy the inclusion criterion (i.e., employed in health or care services) and were excluded. The first COPSOQ item was answered by 9844 nurses while 8784 answered the last question. In total, 8804 answered 85% or more of the questions (more than 112 of all 132 items) and this is the sample used in this initial validation study.

Thirteen of the 99 items in part a) of the questionnaire were omitted because the content was considered to be too similar to other items. The statistical analyses used in this initial validation study thus include 86 of the 99 items in the work environment part of COPSOQ III.

The project was reported to the Norwegian Center for Research Data AS (project no. 731523) and informed consent was obtained from all respondents according to Norwegian and European General Data Protection Regulation.

## Analyses

**Descriptive statistics.** Distributional descriptive statistics of item response and dimension characteristics including floor and ceiling distribution and standard reliability coefficients are calculated. To explore the statistical properties of the data, we use item response theory (IRT).

**Item response theory.** IRT models are general mathematical models that probabilistically describe the relation between a person's response to an item and the latent variable that is unobservable, and its fundament was established when a mathematician discovered that only

two model parameters are sufficient statistics to study test results, that is, item difficulties and examinee ability [50]. Initially, IRT models could be used to study any situation in which a number of subjects perform a series of tasks or answer questions having the same two alternative responses [51]. Later, IRT was developed to cover polytomous items, including the ordered items on which COPSOQ is based. Generalized partial credit models are the most flexible polytomous IRT models because they have fewer assumptions [52]. The actual work environment characteristic or exposure, such as 'Quantitative Demands' and 'Work Pace' are not observed and must be treated as latent variables. The intuition behind the IRT models in our setting is that we can calculate the probability of a person with a certain 'perceived work environment characteristic' responding correctly to an item with given properties. The properties of the items are described as item discrimination and item difficulty. IRT utilizes the explicit mathematical model for the probability of each possible response to an item, and the probability is derived as a function of the latent variable and item parameters and then used to obtain the likelihood of what we aim to measure, that is, the exposure as a function of the observed responses and item parameters. Partial credit models such as IRT can thus be understood as models that link polytomous manifest variables to latent variables.

Let $Y_{ij}$ be the outcome of item $i$ from person $j$ where all items take on the ordered categories, $k = 1, 2, \ldots, K$. IRT shows that the probability of a person $j$ with an unobserved characteristic of their work environment or exposure $\theta_j$ answer response $k$ for item $i$ and is given by:

$$\Pr\left(Y_{ij} = k | a_i, \mathbf{b}_i, \theta_j\right) = \frac{\exp\{\sum_{t=1}^{k} a_i(\theta_j - b_{it})\}}{1 + \sum_{s=1}^{K} \exp\{\sum_{t=1}^{s} a_i(\theta_j - b_{it})\}}$$

where $\alpha_i$ represents the discrimination (how well it differentiates between individuals) for item $i$, while $\mathbf{b}_i = (b_{i1}, \ldots, b_{iK})$ represents the difficulties that distinguish the ordered categories of item $i$, and thus the probability of choosing the first category of item $i$ is:

$$\Pr\left(Y_{ij} = 1 | a_i, \mathbf{b}_i, \theta_j\right) = \frac{1}{1 + \sum_{s=1}^{K} \exp\{\sum_{t=1}^{s} a_i(\theta_j - b_{it})\}}$$

The probability for providing response $k$ is parameterized as:

$$\Pr\left(Y_{ij} = k | \alpha_i, \beta_i, \theta_j\right) = \frac{\exp(k\alpha_i\theta_j + \beta_{ik})}{1 + \sum_{s=1}^{K} \exp(s\alpha_i\theta_j + \beta_{is})}$$

where $b_{ik} = -(\beta_{ik} - \beta_{i,k-1})/\alpha_i$ and $b_{i0} = 0$ and $\beta_{i0} = 0$ and $k$ $\alpha_i$ and $\beta_{ik}$ are item response parameters [53]. The calibration of the dimension is conducted by maximum likelihood estimation involving iteration between the item values across persons and the person values across items. We use both generalized partial credit models and estimate different discrimination parameters $\alpha_i$ for all items in a dimension when possible, however for dimensions that do not reach convergence we restrict the model to only estimate one $\alpha_i$ assuming that the discrimination is equal for all items in a dimension.

The items in part a) of COPSOQ all have five ordered response categories, and thus there are 5−1 = 4 threshold parameters and one unique slope parameter to be estimated for each item. Technically, each threshold reflects the level of general perceived work environment characteristic needed to have equal probability of choosing to respond above a given threshold.

We present the estimated alphas and betas in addition to graphical illustrations of the results using item information functions (IIFs). IIFs indicate the range of difficulty levels where an item is best at discriminating between individuals. The item response function is

thus a mathematical function that relates the latent variable to the probability of responding with each possible answer to an item, and the IIF is an indication of item quality and the item's ability to differentiate between respondents. More information, determined by the item's discrimination parameter, indicates higher accuracy or reliability for measuring a person's latent level of work environment exposure. Item information can be used to select a set of items that together provide much information on a desired range of the latent dimension and is the IRT alternative to the concept of reliability as the sum of functions indicates the amount of information the total set of items conveys for persons with different latent levels of exposure (θ). More information (a higher value on the *y-axis* of the IIF) indicates more precise measurement at the continuum of θ. The value on the *y-axis* shows how much empirical information each item is adding to the dimension. The value on the *x-axis* shows where this empirical information is occurring along the continuum of the latent work environment variables (θ).

**Factor analysis.** Evidence-based practice requires scales with known properties and it is suggested that knowledge of those properties is more complete when researchers use both confirmatory factor analysis (CFA) and IRT [54]. A standard CFA including standard test statistics to evaluate the latent dimensionality or factor structure of scores is therefore included in the S2 Appendix. However, this analysis has limitations as it restricts items to load onto other dimensions than they are assigned to, which may be overly strict and unrealistic [55]. CFA can thus be contrasted with an exploratory factor analysis (EFA) where all loadings are free to vary [56]. The results from EFA using maximum likelihood without any restrictions imposed are included in S2 Appendix to provide a more thorough analysis of separability of dimensions. CFA was performed using the jamovi 2.2.5 software [57] while all other analyses were conducted using Stata/SE 16.1 for Windows (64-bit x86-64).

**Sample adequacy measure.** The Kaiser-Meyer-Olkin measure of sampling adequacy compares the correlations and the partial correlations between items and reflects the proportion of variance among variables that might be common variance. High levels imply high correlation relative to partial correlation and the data are suitable for low-dimensional representation or factor analysis. Values above 0.5 are generally accepted as indicating the adequacy of the sample, while values below 0.5 imply that the sample is inadequate. Values between 0.8 and 1 imply that the data are very suitable for factor analysis [58].

**Goodness of fit.** Results from chi-squared test are provided together with the commonly reported statistic and root mean square error of approximation (RMSEA), which incorporates a penalty function for poor model parsimony where a value of about 0.05 or less would indicate a close fit of the model in relation to the degrees of freedom [59]. We also report the comparative fit index (CFI), which evaluates the fit of one model relative to a more restricted baseline model, and the Tucker–Lewis index (TLI), which measures the relative reduction in misfit per degree of freedom. CFI and TLI values close to 0.95 indicate a good fit of the model [60]. Consistency between items (internal consistency) in a dimension is measured by Cronbach's alpha, and because this is based on restrictive assumptions such as tau-equivalence, we also calculate McDonald's omega to estimate dimension or scale reliability [61].

## Results

### Translation

The translated Norwegian questionnaire includes all 146 original items (see S1 Appendix). Translation problems encountered were most often related to the generalized language used in the original English version. For instance, in item TM4, 'Are the employees able to express their views and feelings?', the wording 'able to' feels too imprecise in Norwegian and sounds as if we are asking if they 'know how to' express their views and feelings. We settled on a

translation closer to the English wording 'is it possible'. Furthermore, because the English expression 'shown up' [in front of others] used in item BU2 does not have a precise translation in Norwegian, what we ended up with might be closer to 'humiliated' than 'shown up'. All original items were translated; however, 14 items were excluded for different reasons in the initial Norwegian translation. For instance, item BO4, 'How often have you felt tired?' was excluded because most people feel tired every night, while item RE2, 'Does the management at your workplace respect you?' was excluded because it is very similar in Norwegian to item RE1, 'Is your work recognized and appreciated by the management?'.

## Descriptive statistics

As seen from Table 1, all response alternatives for all items have observations, although item CD2, 'Does your work require that you remember a lot of things?' only has seven respondents answering "Never/almost never", indicating that most jobs require some memory capacity. However, it is the combination of response alternatives for the items in a dimension that is important, as the presence of a combination without observations will cause convergence problems in the generalized partial credit model and it is necessary to constrain at least one parameter. The non-generalized partial credit model constrains the discrimination parameters to be the same for all items. For instance, there are no observations that have the combination of the first and last response alternatives for item WP1, 'Do you have to work very fast?' and WP2, 'Do you work at a high pace throughout the day?'; that is, nobody answered that they always have to work very fast and at the same time never/almost never have to work at a high pace throughout the day.

Most missing observations are found for item TM3, 'Does the management withhold important information from the employees?' to which 174 persons did not respond (2%). On average, for all 86 items, 26 individuals (0.3%) did not respond.

The strongest ceiling effect is found in the Meaning of Work dimension (items MW1 and MW2) in this sample, and the strongest floor effect in the Job Insecurity (items JI1, JI2 and JI3), implying that the majority of nurses report that their work is meaningful and important, and that few nurses fear for their job.

## Properties of the items

Table 2 shows that low discrimination (low value of parameter a) is found for several items, and these are candidates to be removed because they do not contribute to distinguishing between respondents to any great extent.

Fourteen items are candidates to be removed (0 core items, 1 middle item and 13 long items), see S1 Table in S2 Appendix.

Figs 1–5 include IIFs for all items in all the dimensions, and low discrimination corresponds to a small slope of the IIF. At the top left of Fig 1, the IIFs for items QD1, QD2 and QD3 are shown for the latent or unobserved work environment characteristic or the dimension of Quantitative Demands (θ). This is a particularly good dimension as much of the continuum of the dimension (*x-axis*) is covered by the IIFs. This implies that the three items cover the whole scale of the latent dimension Quantitative Demands. The dimension Cognitive Demands on the other hand, has little coverage at the lower end of the dimension as none of the items discriminate well between respondents at the low part of the *x-axis*, implying that the items in the dimension are better at measuring high levels of exposure than low levels. However, the IIFs of CD3 and CD4 are flat in the figure, and this corresponds to the low value of the discrimination parameter a in Table 2.

**Table 1. Descriptive statistics, item response, N = 8804 (see S1 Appendix for key to items).**

| Item | Alt1 n (%) | Alt 2 n (%) | Alt 3 n (%) | Alt 4 n (%) | Alt 5 n (%) | Total n | Missing n (%) |
|---|---|---|---|---|---|---|---|
| QD1 | 332 (3.8) | 2322 (26.4) | 3936 (44.8) | 1690 (19.2) | 501 (5.7) | 8781 | 23 (0.3) |
| QD2 | 278 (3.2) | 2656 (30.2) | 3454 (39.3) | 1904 (21.7) | 499 (5.7) | 8791 | 13 (0.1) |
| QD3 | 501 (5.7) | 2267 (25.8) | 3754 (42.7) | 1812 (20.6) | 456 (5.2) | 8790 | 14 (0.2) |
| WP1 | 1214 (13.9) | 4180 (47.9) | 2915 (33.4) | 398 (4.6) | 26 (0.3) | 8733 | 71 (0.8) |
| WP2 | 1700 (19.3) | 3599 (40.9) | 2759 (31.4) | 646 (7.3) | 91 (1) | 8795 | 9 (0.1) |
| CD1 | 2569 (29.2) | 3593 (40.9) | 1862 (21.2) | 698 (7.9) | 61 (0.7) | 8783 | 21 (0.2) |
| CD2 | 3937 (44.8) | 3730 (42.4) | 952 (10.8) | 168 (1.9) | 7 (0.1) | 8794 | 10 (0.1) |
| CD3 | 357 (4.1) | 2721 (30.9) | 4338 (49.3) | 1301 (14.8) | 75 (0.9) | 8792 | 12 (0.1) |
| CD4 | 517 (5.9) | 3052 (34.7) | 3925 (44.6) | 1236 (14.1) | 66 (0.8) | 8796 | 8 (0.1) |
| ED1 | 600 (6.8) | 4031 (45.9) | 3530 (40.2) | 561 (6.4) | 62 (0.7) | 8784 | 20 (0.2) |
| EDX2 | 1586 (18) | 3764 (42.8) | 2383 (27.1) | 803 (9.1) | 254 (2.9) | 8790 | 14 (0.2) |
| ED3 | 1761 (20.1) | 3036 (34.6) | 3193 (36.4) | 688 (7.8) | 104 (1.2) | 8782 | 22 (0.2) |
| HE1 | 2201 (25.3) | 2087 (24) | 2200 (25.3) | 1676 (19.3) | 538 (6.2) | 8702 | 102 (1.2) |
| HE2 | 698 (7.9) | 2142 (24.4) | 4289 (48.8) | 1302 (14.8) | 355 (4) | 8786 | 18 (0.2) |
| HE3 | 2527 (28.8) | 4142 (47.1) | 1647 (18.7) | 376 (4.3) | 93 (1.1) | 8785 | 19 (0.2) |
| HE4 | 143 (1.6) | 1513 (17.2) | 3668 (41.7) | 2821 (32.1) | 646 (7.3) | 8791 | 13 (0.1) |
| INX1 | 261 (3) | 2376 (27) | 4026 (45.8) | 1771 (20.2) | 354 (4) | 8788 | 16 (0.2) |
| IN2 | 160 (1.8) | 1024 (11.7) | 2269 (25.9) | 2981 (34) | 2343 (26.7) | 8777 | 27 (0.3) |
| IN3 | 75 (0.9) | 741 (8.4) | 3284 (37.4) | 3249 (37) | 1432 (16.3) | 8781 | 23 (0.3) |
| IN4 | 215 (2.4) | 1905 (21.7) | 3948 (44.9) | 2207 (25.1) | 515 (5.9) | 8790 | 14 (0.2) |
| PD2 | 1330 (15.1) | 3335 (37.9) | 3275 (37.3) | 712 (8.1) | 137 (1.6) | 8789 | 15 (0.2) |
| PD3 | 4012 (45.6) | 3815 (43.4) | 843 (9.6) | 98 (1.1) | 23 (0.3) | 8791 | 13 (0.1) |
| PD4 | 1142 (13) | 3378 (38.4) | 3202 (36.4) | 861 (9.8) | 206 (2.3) | 8789 | 15 (0.2) |
| VA1 | 3154 (35.8) | 4085 (46.4) | 1308 (14.9) | 220 (2.5) | 34 (0.4) | 8801 | 3 (0) |
| VA2r | 49 (0.6) | 570 (6.5) | 2604 (29.6) | 4916 (55.9) | 653 (7.4) | 8792 | 12 (0.1) |
| CT1 | 541 (6.2) | 2771 (31.5) | 2804 (31.9) | 1752 (19.9) | 927 (10.5) | 8795 | 9 (0.1) |
| CT2 | 471 (5.4) | 2848 (32.4) | 2709 (30.8) | 1789 (20.4) | 965 (11) | 8782 | 22 (0.2) |
| CT3 | 272 (3.1) | 2048 (23.3) | 3740 (42.5) | 1959 (22.3) | 773 (8.8) | 8792 | 12 (0.1) |
| CT4 | 506 (5.8) | 880 (10) | 1098 (12.5) | 1194 (13.6) | 5101 (58.1) | 8779 | 25 (0.3) |
| CT5r | 621 (7.1) | 2003 (22.8) | 3967 (45.1) | 2099 (23.9) | 104 (1.2) | 8794 | 10 (0.1) |
| MW1 | 4023 (45.9) | 3819 (43.5) | 819 (9.3) | 89 (1) | 23 (0.3) | 8773 | 31 (0.4) |
| MW2 | 4671 (53.2) | 3447 (39.2) | 588 (6.7) | 63 (0.7) | 19 (0.2) | 8788 | 16 (0.2) |
| PR1 | 407 (4.6) | 2288 (26) | 3815 (43.4) | 1681 (19.1) | 597 (6.8) | 8788 | 16 (0.2) |
| PR2 | 678 (7.7) | 4427 (50.4) | 3149 (35.8) | 437 (5) | 99 (1.1) | 8790 | 14 (0.2) |
| RE1 | 964 (11) | 3151 (35.8) | 3000 (34.1) | 1182 (13.4) | 493 (5.6) | 8790 | 14 (0.2) |
| RE3 | 1543 (17.5) | 4458 (50.7) | 2141 (24.3) | 454 (5.2) | 200 (2.3) | 8796 | 8 (0.1) |
| CL1 | 1928 (21.9) | 4916 (55.9) | 1675 (19) | 222 (2.5) | 53 (0.6) | 8794 | 10 (0.1) |
| CL2 | 2748 (31.3) | 4818 (54.8) | 1041 (11.8) | 151 (1.7) | 29 (0.3) | 8787 | 17 (0.2) |
| CL3 | 2534 (28.8) | 5097 (58) | 1012 (11.5) | 124 (1.4) | 28 (0.3) | 8795 | 9 (0.1) |
| CO2 | 272 (3.1) | 947 (10.9) | 3445 (39.6) | 3128 (35.9) | 912 (10.5) | 8704 | 100 (1.1) |
| CO3 | 372 (4.2) | 1531 (17.4) | 4748 (54.1) | 1862 (21.2) | 268 (3.1) | 8781 | 23 (0.3) |
| IT | 790 (9) | 1689 (19.2) | 4122 (46.9) | 1873 (21.3) | 311 (3.5) | 8785 | 19 (0.2) |
| QLX1 | 749 (8.5) | 2961 (33.7) | 3392 (38.6) | 1241 (14.1) | 435 (5) | 8778 | 26 (0.3) |
| QL3 | 783 (9) | 3401 (38.9) | 3193 (36.5) | 976 (11.2) | 388 (4.4) | 8741 | 63 (0.7) |
| QL4 | 739 (8.5) | 2804 (32.1) | 3209 (36.7) | 1333 (15.2) | 660 (7.5) | 8745 | 59 (0.7) |
| SSX1 | 2682 (30.5) | 3158 (36) | 2006 (22.8) | 684 (7.8) | 252 (2.9) | 8782 | 22 (0.2) |
| SSX2 | 2028 (23.1) | 3230 (36.8) | 2345 (26.7) | 868 (9.9) | 314 (3.6) | 8785 | 19 (0.2) |

*(Continued)*

**Table 1.** (Continued)

| Item | Alt1 n (%) | Alt 2 n (%) | Alt 3 n (%) | Alt 4 n (%) | Alt 5 n (%) | Total n | Missing n (%) |
|---|---|---|---|---|---|---|---|
| SCX1 | 3162 (36) | 4494 (51.2) | 1027 (11.7) | 94 (1.1) | 8 (0.1) | 8785 | 19 (0.2) |
| SCX2 | 3340 (38) | 4147 (47.2) | 1115 (12.7) | 163 (1.9) | 21 (0.2) | 8786 | 18 (0.2) |
| SW1 | 3563 (40.5) | 4719 (53.7) | 472 (5.4) | 24 (0.3) | 9 (0.1) | 8787 | 17 (0.2) |
| SW2 | 2456 (27.9) | 5458 (62.1) | 804 (9.1) | 62 (0.7) | 9 (0.1) | 8789 | 15 (0.2) |
| SW3 | 3956 (45.1) | 3547 (40.4) | 999 (11.4) | 237 (2.7) | 42 (0.5) | 8781 | 23 (0.3) |
| CWX3 | 1751 (19.9) | 3421 (38.9) | 2463 (28) | 756 (8.6) | 400 (4.6) | 8791 | 13 (0.1) |
| CW4r | 2067 (23.6) | 1973 (22.5) | 2461 (28) | 1806 (20.6) | 470 (5.4) | 8777 | 27 (0.3) |
| CW5 | 2063 (23.5) | 3782 (43) | 2258 (25.7) | 495 (5.6) | 199 (2.3) | 8797 | 7 (0.1) |
| WE1 | 310 (3.5) | 3987 (45.3) | 3462 (39.4) | 938 (10.7) | 98 (1.1) | 8795 | 9 (0.1) |
| WE2 | 2138 (24.3) | 4986 (56.7) | 1446 (16.4) | 194 (2.2) | 28 (0.3) | 8792 | 12 (0.1) |
| WE3 | 798 (9.1) | 4035 (45.9) | 3005 (34.2) | 845 (9.6) | 103 (1.2) | 8786 | 18 (0.2) |
| JI1 | 68 (0.8) | 92 (1) | 322 (3.7) | 1442 (16.4) | 6877 (78.1) | 8801 | 3 (0) |
| JI2 | 27 (0.3) | 72 (0.8) | 346 (3.9) | 2311 (26.3) | 6036 (68.7) | 8792 | 12 (0.1) |
| JI3 | 173 (2) | 313 (3.6) | 954 (10.9) | 2321 (26.4) | 5030 (57.2) | 8791 | 13 (0.1) |
| IW1 | 335 (3.8) | 404 (4.6) | 1268 (14.4) | 2658 (30.2) | 4127 (46.9) | 8792 | 12 (0.1) |
| IW2 | 387 (4.4) | 781 (8.9) | 2319 (26.4) | 2998 (34.1) | 2314 (26.3) | 8799 | 5 (0.1) |
| IW3 | 483 (5.5) | 759 (8.6) | 1886 (21.5) | 2705 (30.8) | 2952 (33.6) | 8785 | 19 (0.2) |
| IW4 | 712 (8.1) | 715 (8.1) | 1433 (16.3) | 2809 (32) | 3121 (35.5) | 8790 | 14 (0.2) |
| IW5 | 1023 (11.7) | 2914 (33.3) | 3223 (36.8) | 1118 (12.8) | 478 (5.5) | 8756 | 48 (0.5) |
| QW1 | 622 (7.1) | 4673 (53.2) | 3023 (34.4) | 398 (4.5) | 66 (0.8) | 8782 | 22 (0.2) |
| QW2 | 1170 (13.3) | 5091 (57.9) | 2154 (24.5) | 301 (3.4) | 71 (0.8) | 8787 | 17 (0.2) |
| JS1 | 1180 (13.4) | 3960 (45.1) | 2848 (32.5) | 601 (6.8) | 187 (2.1) | 8776 | 28 (0.3) |
| JS2 | 787 (9) | 3796 (43.4) | 2119 (24.2) | 1671 (19.1) | 381 (4.4) | 8754 | 50 (0.6) |
| JS3 | 974 (11.1) | 5076 (57.9) | 1802 (20.5) | 773 (8.8) | 148 (1.7) | 8773 | 31 (0.4) |
| JS4 | 1715 (19.5) | 5175 (58.9) | 1382 (15.7) | 427 (4.9) | 81 (0.9) | 8780 | 24 (0.3) |
| JS5 | 121 (1.4) | 1614 (18.4) | 2027 (23.1) | 3422 (39) | 1583 (18.1) | 8767 | 37 (0.4) |
| WF2 | 1384 (15.8) | 1824 (20.8) | 3340 (38) | 1676 (19.1) | 561 (6.4) | 8785 | 19 (0.2) |
| WF3 | 845 (9.6) | 1373 (15.7) | 3044 (34.7) | 2637 (30.1) | 873 (10) | 8772 | 32 (0.4) |
| WF5 | 638 (7.3) | 1136 (13) | 3016 (34.4) | 2858 (32.6) | 1124 (12.8) | 8772 | 32 (0.4) |
| TE1r | 2040 (23.4) | 4673 (53.7) | 1744 (20) | 208 (2.4) | 41 (0.5) | 8706 | 98 (1.1) |
| TE2r | 1881 (21.7) | 4396 (50.7) | 2022 (23.3) | 303 (3.5) | 74 (0.9) | 8676 | 128 (1.5) |
| TE3 | 1502 (17.1) | 5515 (62.9) | 1606 (18.3) | 133 (1.5) | 18 (0.2) | 8774 | 30 (0.3) |
| TM1 | 2155 (24.5) | 5050 (57.5) | 1335 (15.2) | 196 (2.2) | 46 (0.5) | 8782 | 22 (0.2) |
| TMX2 | 1626 (18.5) | 4514 (51.3) | 2187 (24.9) | 367 (4.2) | 97 (1.1) | 8791 | 13 (0.1) |
| TM3r | 1422 (16.5) | 3834 (44.4) | 2641 (30.6) | 564 (6.5) | 169 (2) | 8630 | 174 (2) |
| TM4 | 1377 (15.7) | 4199 (47.8) | 2552 (29) | 526 (6) | 135 (1.5) | 8789 | 15 (0.2) |
| JU1 | 711 (8.2) | 3649 (41.9) | 3331 (38.3) | 759 (8.7) | 254 (2.9) | 8704 | 100 (1.1) |
| JU2 | 965 (11) | 2880 (32.8) | 3182 (36.2) | 1289 (14.7) | 472 (5.4) | 8788 | 16 (0.2) |
| JU4 | 501 (5.7) | 4002 (45.6) | 3499 (39.8) | 642 (7.3) | 138 (1.6) | 8782 | 22 (0.2) |

Items that do not discriminate well between persons in the sample—those with a small slope, thus appearing relatively flat and at the same time not contributing to covering the range or continuum of exposure not covered by other items in the dimension—are identified as candidates to be removed to reduce the length of the questionnaire. However, it would be premature to remove these items based on analyses of one relatively homogeneous sample, as other samples might show other statistical properties and could give other results and thus provide other suggestions for candidates.

**Table 2. Discrimination and difficulty parameters derived from generalized partial credit models.**

| | | Discrimination | Step Parameters | | | |
|---|---|---|---|---|---|---|
| | | a | b1 | b2 | b3 | b4 |
| QD | QD1 | 1.59 | −2.41 | −0.68 | 0.95 | 1.92 |
| | QD2 | 3.03 | −2.15 | −0.46 | 0.68 | 1.75 |
| | QD3 | 3.81 | −1.73 | −0.52 | 0.71 | 1.75 |
| WP* | WP1 | 3.06 | −1.24 | 0.35 | 1.89 | 3.07 |
| | WP2 | 3.06 | −0.97 | 0.30 | 1.57 | 2.55 |
| CD | CD1 | 1.67 | −0.63 | 0.74 | 1.62 | 3.17 |
| | CD2 | 5.54 | −0.11 | 1.19 | 2.11 | 3.29 |
| | CD3** | 0.55 | −4.30 | −0.97 | 2.54 | 5.99 |
| | CD4** | 0.76 | −3.05 | −0.43 | 2.00 | 4.90 |
| ED | ED1 | 2.64 | −1.77 | 0.08 | 1.75 | 2.81 |
| | EDX2** | 0.93 | −1.46 | 0.59 | 1.86 | 2.49 |
| | ED3 | 5.11 | −0.89 | 0.11 | 1.42 | 2.39 |
| HE | HE1** | 0.54 | −0.33 | −0.12 | 0.88 | 2.88 |
| | HE2 | 1.37 | −1.81 | −0.74 | 1.37 | 2.17 |
| | HE3 | 0.94 | −0.85 | 1.30 | 2.50 | 3.02 |
| | HE4 | 1.17 | −3.29 | −1.29 | 0.39 | 2.13 |
| IN | INX1 | 1.45 | −2.68 | −0.71 | 1.00 | 2.31 |
| | IN2** | 0.59 | −4.16 | −1.91 | −0.57 | 0.76 |
| | IN3 | 1.16 | −3.44 | −2.05 | −0.06 | 1.34 |
| | IN4 | 3.27 | −2.21 | −0.79 | 0.56 | 1.74 |
| PD | PD2 | 1.77 | −1.28 | 0.09 | 1.72 | 2.55 |
| | PD3 | 0.95 | −0.05 | 2.17 | 3.49 | 3.37 |
| | PD4 | 3.62 | −1.24 | 0.05 | 1.29 | 2.12 |
| VA* | VA1 | 0.83 | 0.53 | −1.78 | −3.16 | −3.87 |
| | VA2r | 0.83 | 4.27 | 2.54 | 0.86 | −2.99 |
| CT | CT1 | 1.44 | −2.17 | −0.28 | 0.70 | 1.42 |
| | CT2** | 0.66 | −3.47 | −0.14 | 0.88 | 1.64 |
| | CT3 | 1.66 | −2.52 | −0.80 | 0.71 | 1.58 |
| | CT4 | 1.01 | −1.85 | −1.04 | −0.42 | −1.25 |
| | CT5r** | 0.28 | −4.52 | −2.52 | 2.42 | 11.06 |
| MW* | MW1 | 2.50 | −0.10 | 1.50 | 2.64 | 3.00 |
| | MW2 | 2.50 | 0.12 | 1.72 | 2.77 | 3.04 |
| PR | PR1 | 3.64 | −1.87 | −0.55 | 0.73 | 1.59 |
| | PR2 | 1.42 | −2.12 | 0.33 | 2.29 | 2.69 |
| RE | RE1 | 2.88 | −1.41 | −0.07 | 1.02 | 1.70 |
| | RE3 | 2.23 | −1.14 | 0.62 | 1.76 | 2.09 |
| CL | CL1 | 1.38 | −1.12 | 1.18 | 2.63 | 2.89 |
| | CL2 | 2.68 | −0.56 | 1.30 | 2.33 | 2.95 |
| | CL3 | 3.20 | −0.62 | 1.27 | 2.36 | 2.88 |
| CO | CO2 | 2.27 | −2.09 | −1.34 | 0.11 | 1.50 |
| | CO3 | 1.00 | −2.44 | −1.49 | 1.27 | 2.97 |
| QL | QLX1 | 2.16 | −1.66 | −0.21 | 1.08 | 1.83 |
| | QL3 | 3.15 | −1.52 | −0.04 | 1.14 | 1.80 |
| | QL4 | 2.23 | −1.65 | −0.26 | 0.93 | 1.54 |
| SS* | SSX1 | 5.31 | −0.46 | 0.48 | 1.31 | 1.97 |
| | SSX2 | 5.31 | −0.69 | 0.30 | 1.17 | 1.87 |

*(Continued)*

**Table 2.** (Continued)

| | | Discrimination | Step Parameters | | | |
|---|---|---|---|---|---|---|
| | | a | b1 | b2 | b3 | b4 |
| SC* | SCX1 | 3.08 | −0.39 | 1.31 | 2.56 | 3.37 |
| | SCX2 | 3.08 | −0.33 | 1.20 | 2.28 | 3.04 |
| SW | SW1 | 3.44 | −0.26 | 1.79 | 3.01 | 3.10 |
| | SW2 | 3.28 | −0.65 | 1.47 | 2.71 | 3.27 |
| | SW3 | 2.53 | −0.12 | 1.27 | 2.12 | 2.89 |
| CW | CWX3 | 4.71 | −0.88 | 0.24 | 1.21 | 1.75 |
| | CW4 | −1.00 | 2.38 | 0.79 | −0.27 | −0.56 |
| | CW5 | 2.05 | −0.86 | 0.54 | 1.78 | 2.16 |
| WE* | WE1 | 1.59 | −2.62 | 0.02 | 1.58 | 3.00 |
| | WE2 | 1.59 | −0.95 | 1.26 | 2.57 | 3.32 |
| | WE3 | 1.59 | −1.84 | 0.23 | 1.62 | 2.93 |
| JI | JI1 | 2.77 | −2.17 | −2.15 | −1.80 | −0.98 |
| | JI2 | 1.38 | −2.65 | −2.65 | −2.34 | −0.86 |
| | JI3 | 1.42 | −2.01 | −1.98 | −1.30 | −0.50 |
| IW | IW1 | 1.51 | −1.61 | −1.75 | −0.98 | −0.11 |
| | IW2 | 1.27 | −1.81 | −1.58 | −0.39 | 0.70 |
| | IW3 | 1.85 | −1.58 | −1.30 | −0.48 | 0.37 |
| | IW4** | 0.94 | −1.09 | −1.40 | −0.93 | 0.23 |
| | IW5** | −0.33 | 3.53 | 0.35 | −3.46 | −3.11 |
| QW | QW1 | 2.05 | −1.90 | 0.35 | 2.07 | 2.80 |
| | QW2 | 1.77 | −1.50 | 0.79 | 2.27 | 2.73 |
| JS | JS1 | 1.92 | −1.40 | 0.31 | 1.73 | 2.21 |
| | JS2** | 0.77 | −2.67 | 0.69 | 0.76 | 2.86 |
| | JS3 | 1.85 | −1.64 | 0.78 | 1.43 | 2.50 |
| | JS4 | 3.90 | −0.93 | 0.89 | 1.66 | 2.45 |
| | JS5** | 0.42 | −6.98 | −0.92 | −1.27 | 2.20 |
| WF | WF2 | 3.38 | −1.02 | −0.39 | 0.71 | 1.65 |
| | WF3 | 4.03 | −1.35 | −0.70 | 0.26 | 1.35 |
| | WF5 | 3.04 | −1.54 | −0.94 | 0.13 | 1.23 |
| TE* | TE1r | 1.69 | −0.96 | 1.03 | 2.48 | 2.83 |
| | TE2r | 1.69 | −1.03 | 0.83 | 2.25 | 2.59 |
| | TE3 | 1.69 | −1.32 | 1.19 | 2.78 | 3.18 |
| TM | TM1 | 1.73 | −0.91 | 1.29 | 2.41 | 2.77 |
| | TMX2 | 4.54 | −0.95 | 0.57 | 1.71 | 2.29 |
| | TM3r** | 1.25 | −1.40 | 0.45 | 2.06 | 2.39 |
| | TM4** | 1.30 | −1.47 | 0.57 | 2.09 | 2.54 |
| JU | JU1 | 2.75 | −1.65 | 0.03 | 1.41 | 1.99 |
| | JU2 | 2.04 | −1.49 | −0.16 | 1.06 | 1.78 |
| | JU4 | 1.23 | −2.54 | 0.11 | 2.12 | 2.65 |

\* Partial credit models (discrimination parameters constrained to be equal for all items in the dimension).

\*\* Candidates to be removed based on this sample.

In the dimension Control over Working Time shown in the top right of Fig 2, the items CT4 and CT5r are below and inside the range of the other items IIFs and do not contribute much to the measurement of the dimension.

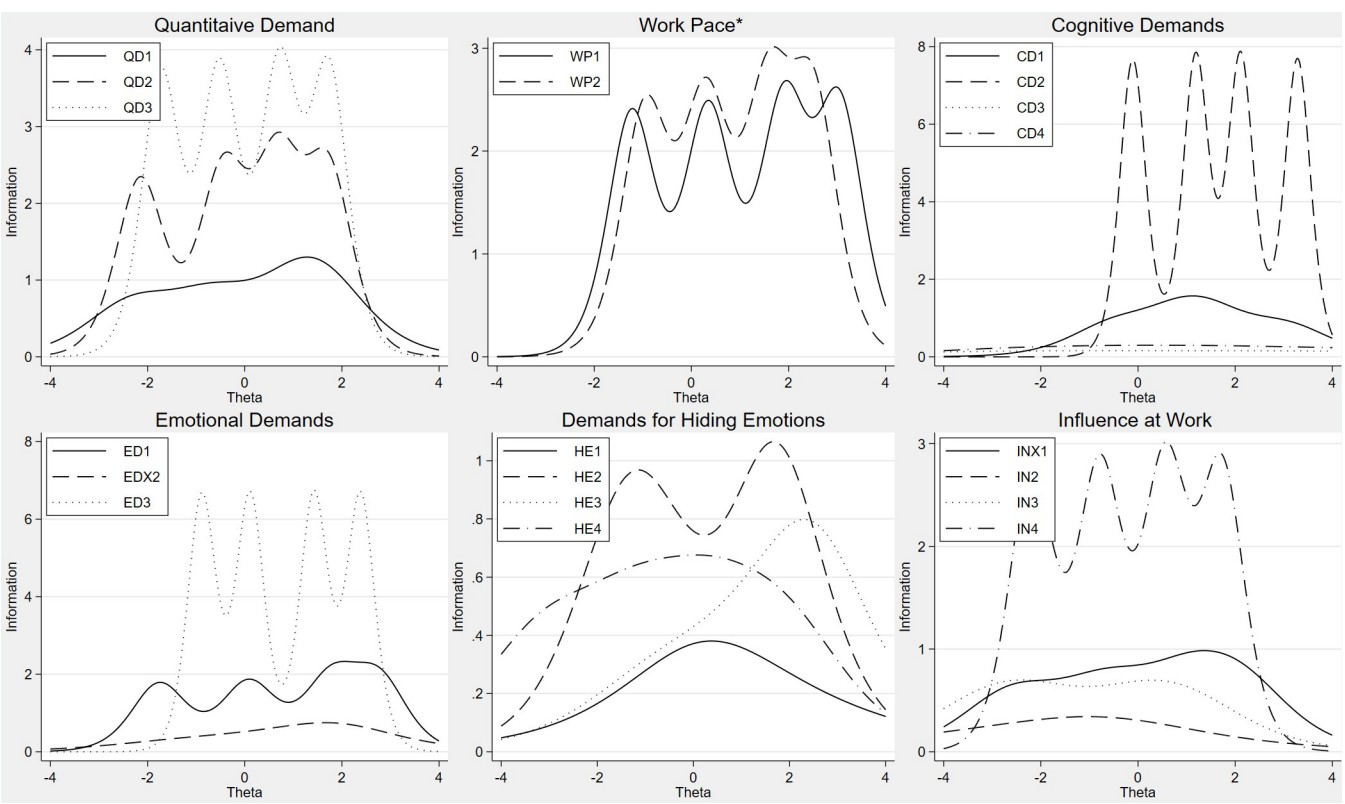

**Fig 1. Item information functions from generalized partial credit models or partial credit models (*), QD, WP, CD, ED, HE, IN.**

Fig 3 shows that the dimension Social Support from Superior has low coverage at both low and high support, while the two next dimensions Social Support from Colleagues and Sense of Community at Work are better at measuring high levels than low levels.

Fig 4 shows for instance that items JS2 and JS5 contribute little to the measurement of Job Satisfaction, see also corresponding low values of discrimination in Table 2. The dimension Vertical Trust in Fig 5 has little contribution from items TM3 and TM4 and are candidates to be removed based on this sample. Analyses of other samples can suggest other candidates.

## Latent dimensionality

The Kaiser–Meyer–Olkin measure of sampling adequacy shows that the correlations between items are in general high, and thus the data are well suited for both IRT and factor analyses (see S1 Table in S2 Appendix).

The results from the CFA show that all items load onto the dimensions they are meant to measure with high statistical precision (see S2 Table in S2 Appendix). Furthermore, including all 86 items in an unrestricted factor analysis using the maximum likelihood method returns nine factors with latent roots or eigenvalues greater than one. However, we retain 30 factors to explore the structure of the latent variables and identify dimensions that load onto the same factors to a high degree (see S3 Table in S2 Appendix). By inspecting the factor loading patterns, we can identify the following dimensions that mostly load onto one factor: QD, WP, CD (CD3 and CD4 have low factor loadings), ED, HE, IN (IN2 low), PD (PD3 low), CT (CT5 low), MW, CL, WE, JI, IW (IW5 low), WF, TE (TE3 low), while PR, RE, QL, SS, CW, JS, TM and JU all have high loadings onto the same factor in this sample, implying that the items measure the same underlying latent dimension. However, this may not be true for other samples.

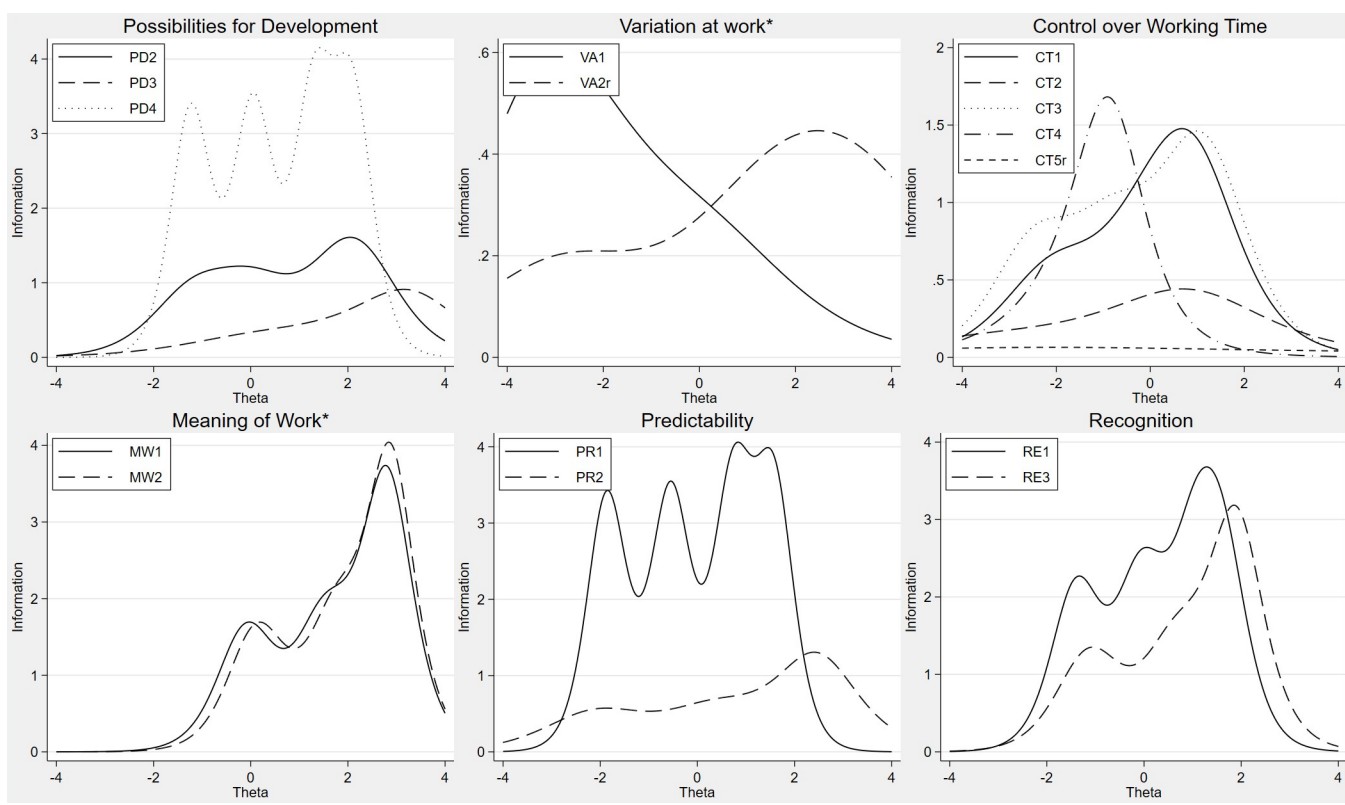

**Fig 2. Item information functions from generalized partial credit models or partial credit models (*), PD, VA, CT, MW, PR, RE.**

We also find that the items CO3 and IT1 seem to form a common factor, suggesting that these two items measure a unique dimension. These results confirm the IRT results as expected items with low values on the discrimination parameters also have low factor loading. Overall, the results confirm the scale structure in the original English version.

## Discussion

### Motive for the translation

Monitoring working conditions to protect employees from work environment hazards relates to human rights covered by laws and regulations, moral obligations, and effective use of the available human capital. Increasing globalization and associated migrant labour challenge the job quality in the part of the labour market characterized by low wages and unskilled workers. For instance, precarious employment is currently a major concern in Europe's labour markets and workers' rights are under pressure [62]. However, work environment factors—for example, exposure to high workload, high work pace and other factors that cause work related illness and sickness absenteeism are also frequently found in workplaces situated in countries with well-developed and robust labour relations, strong employment protection and active labour market policies such as the Nordic countries [63–65].

### Empirical results

The main empirical results are based on IRT models for graded response items. The results show that the structure and the latent dimensionality in the original English version are

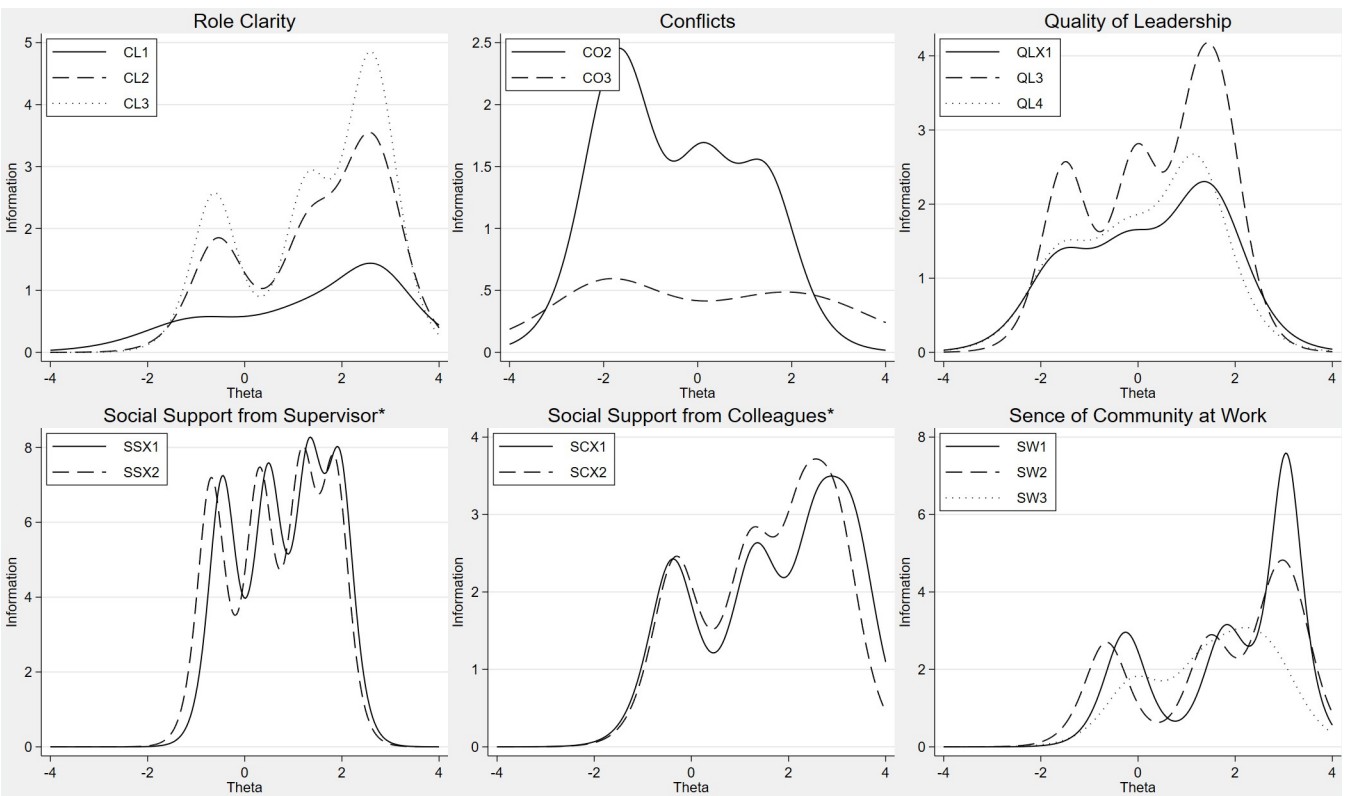

**Fig 3. Item information functions from generalized partial credit models or partial credit models (*), CL, CO, QL, SS, SC, SW.**

replicated in the Norwegian version on the sample of 8804 registered nurses. The scale reliability was high for all dimensions except 'Demands for Hiding Emotions'. The results also show that eight of the 28 dimensions are probably not separate dimensions as items covering these dimensions loaded onto the same factor. This might be a consequence of the homogeneous sample of registered nurses working in health and care services. There may be insufficient variation between these dimensions in this sample to differentiate the type of exposure measured. Furthermore, items in some dimensions provide little information at high and low exposure, and this can also be a feature caused by a homogeneous sample rather than by the instrument. With the exceptions of the items EDX2, 'Do you have to deal with other people's personal problems as part of your work?' and PD3, 'Can you use your skills or expertise in your work?', the core items discriminate well between respondents. Of the 133 items included, 14 were found to be candidates to be removed to obtain a shorter version.

## Other COPSOQ studies confirming structural validity

For COPSOQ II, numerous dimensions were divided into seven overarching domains [66] and in the updated COPSOQ III, this overarching structure was maintained with a few changes [67]. In recent years, the structural validity of different-length versions of COPSOQ II and III has been investigated by confirmatory approaches in diverse populations and countries.

Dicke and colleagues developed a novel approach, called set exploratory structural equation modelling (set-ESEM), where cross-loadings were only allowed within a priori defined sets of factors, and they found support for the suggested structure of COPSOQ II among Australian

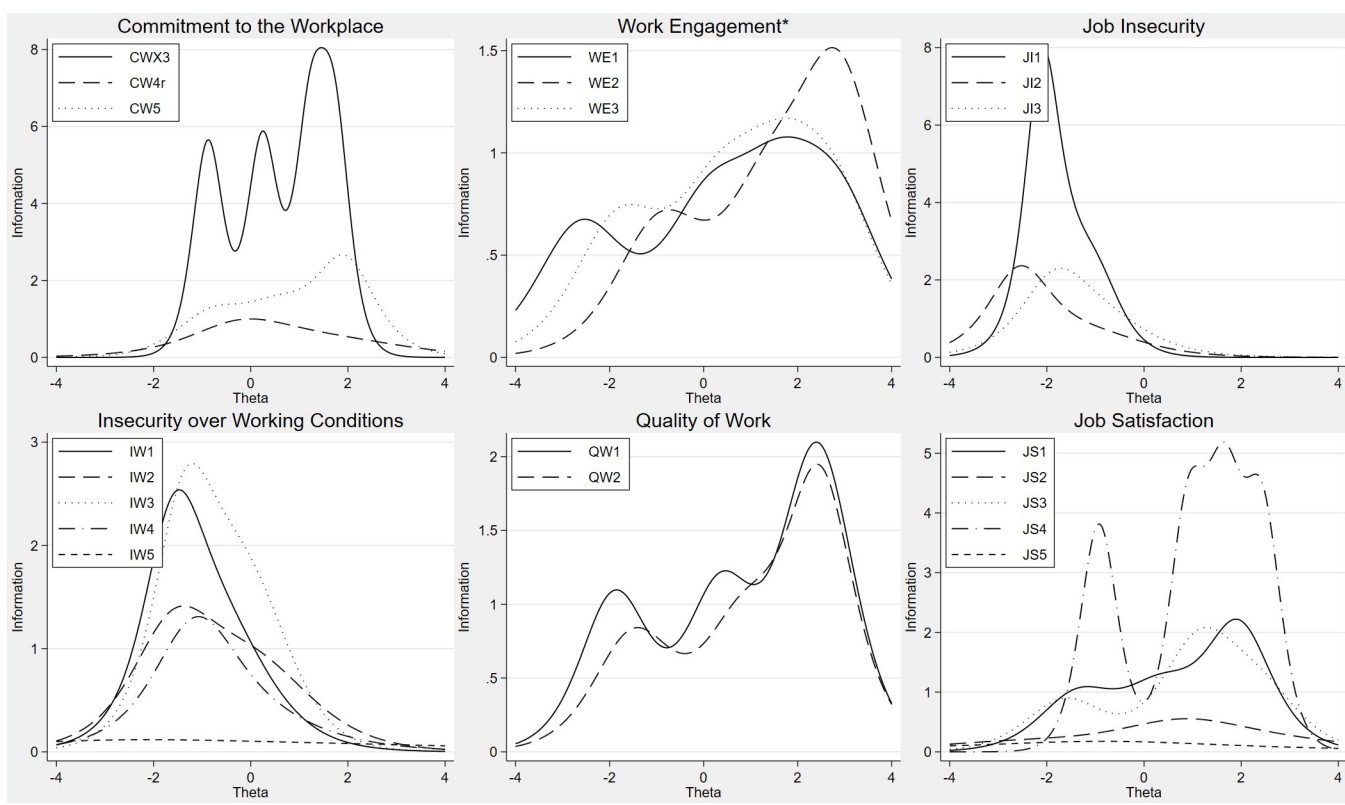

**Fig 4. Item information functions from generalized partial credit models or partial credit models (\*), CW, WE, JI, IW, QW, JS.**

school principals [68]. On the other hand, findings from other studies may question the suggested dimensions. For example, a short version of COPSOQ II was validated among administrative health-care staff from Iran [69]. Based on an EFA, four domains were defined and corroborated by a CFA [67]. However, these domains did not correspond with the original suggested structure of the instrument. Using a corresponding approach, eight factors were identified in a broad sample from Germany [70], and the German standard version of COP-SOQ III operates with five overarching domains as in previous versions [71]. Finally, a Swedish study found support for alternative domains based on theoretical reasoning testing an extended Job Demands–Resources model with three kinds of resources (leadership, task and interpersonal), demands, strain symptoms, and positive work attitudes [72].

The construct validity of COPSOQ II has been supported on the dimension level by using single items as indicators, and on the domain level by using the mean values of scales as indicators among workers from the gas and oil industry in Egypt [73] and in relation to a mix of COPSOQ II and III among French–Italian health-care workers from Switzerland [74]. Second-order models for COPSOQ II have been tested in a sample of health-care workers from China [75] and in a diverse sample from Peru [76]. Furthermore, the structural validity of suggested COPSOQ II dimensions is supported in a model including all originally proposed dimensions as latent variables indicated by items for Polish human service workers [77]. In line with this, a Canadian population study found support for dimensions in a short version comprising selected COPSOQ II and III items [78]. For COPSOQ III, similar approaches have provided general support for dimensions in diverse samples from Turkey [79], the Netherlands [80] and Portugal [81].

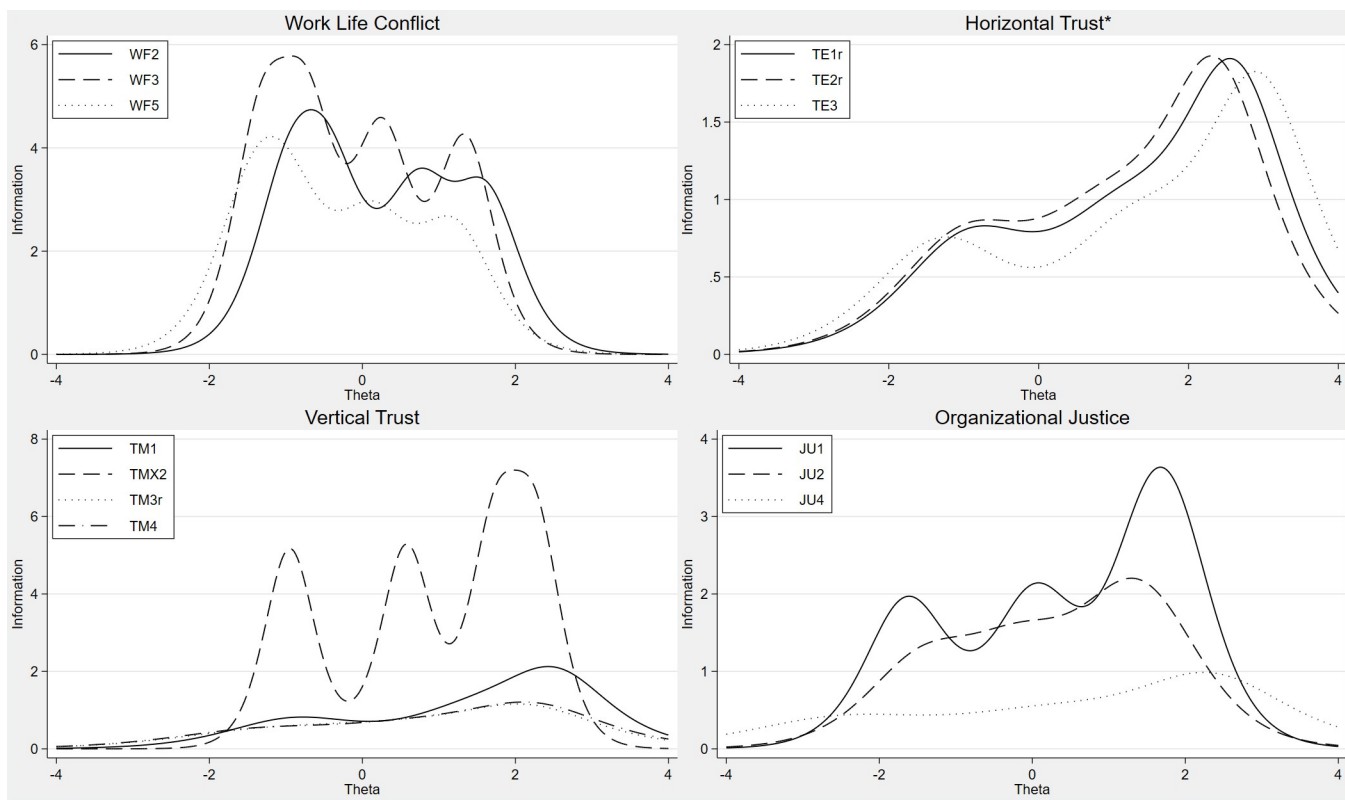

**Fig 5. Item information functions from generalized partial credit models or partial credit models (\*), WF, TE, TM, JU.**

## Shortage of labour and investing in human capital

The expected shortage of labour due to an aging population and fewer people of working age [82] might be mitigated by OHS practices that effectively prevent work-related health problems. Workplaces that can offer work environments that provide health benefits rather than impose health problems on employees will likely have a better chance of retaining and recruiting employees in the future. Investing in human capital through improvements in the work environment can turn out to be profitable in the long run.

## Strength and limitations

The main strength of this study is the large national sample of data, covering registered nurses working in primary and secondary health care. The limitation is that the results are not representative for the entire working population, only for registered nurses.

Another strength is that the translation was performed thorough a forward–backward translation process by a multidisciplinary team of researchers in collaboration with professional translators and an experienced linguist. The multidisciplinary team behind the translated version includes competence in labour economics, occupational medicine, physical rehabilitation medicine, organisational psychology, and COPSOQ expert competence.

## Further research

OHS researchers must in general understand the implications of the current development in organisational design, technological job displacement and new work arrangements on the

well-being of the workforce [83], and for nurses in particular, the global nursing shortage require increased attention to their working and employment conditions [84]. More knowledge of prevention of exposure to organisational, psychological, emotional and relational risk factors among employees in health and care services is warranted.

The present study is the first validation study of the COPSOQ III instrument for use in the Norwegian context. However, validation is an ongoing process and future studies are needed for validation of the parts of the questionnaire not included here, using more diverse samples, investigation of other aspects of validity such as predictive validity and test-retest validity. Finally, we suggest that future research challenge the term "psychosocial work environment" by increased focus on contextual factors, which have a greater potential for designing workplace interventions that promote occupational health compared to individual factors.

## Conclusion

The established Norwegian translation of COPSOQ III can be used in further research about working environment factors and health and wellbeing in Norway. The extended use of the instrument internationally enables comparative studies, which can increase the knowledge and understanding of similarities and differences between labour markets in different countries.

This first validation study shows that the Norwegian version has strong statistical properties like the original, and can be used to assess work environment, including relational and emotional risk factors and physical and social resources available at the workplace.

## Supporting information

**S1 Appendix. The translated items.**
(DOCX)

**S2 Appendix. Results from factor analyses.** S1 Table. Kaiser–Meyer–Olkin measure of sampling adequacy. S2 Table. Confirmatory factor analysis. S3 Table. Exploratory factor analysis, maximum likelihood and factor loadings.
(DOCX)

## Author Contributions

**Conceptualization:** Solveig Osborg Ose, Signe Lohmann-Lafrenz, Vilde Hoff Bernstrøm, Hanne Berthelsen, Gunn Hege Marchand.

**Data curation:** Solveig Osborg Ose.

**Formal analysis:** Solveig Osborg Ose.

**Funding acquisition:** Solveig Osborg Ose.

**Investigation:** Signe Lohmann-Lafrenz, Hanne Berthelsen, Gunn Hege Marchand.

**Methodology:** Solveig Osborg Ose, Signe Lohmann-Lafrenz, Vilde Hoff Bernstrøm, Gunn Hege Marchand.

**Project administration:** Signe Lohmann-Lafrenz.

**Software:** Solveig Osborg Ose.

**Supervision:** Hanne Berthelsen, Gunn Hege Marchand.

**Validation:** Solveig Osborg Ose, Signe Lohmann-Lafrenz, Vilde Hoff Bernstrøm, Hanne Berthelsen, Gunn Hege Marchand.

**Visualization:** Solveig Osborg Ose.

**Writing – original draft:** Solveig Osborg Ose.

**Writing – review & editing:** Signe Lohmann-Lafrenz, Vilde Hoff Bernstrøm, Hanne Berthelsen, Gunn Hege Marchand.

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
