## [Decision Letter · Decision Letter 0]

16 Mar 2023

PONE-D-22-27893The Norwegian version of the Copenhagen Psychosocial Questionnaire (COPSOQ III): Initial validation study using a national sample of licensed nursesPLOS ONE

Dear Dr. Ose,

Thank you for submitting your manuscript to PLOS ONE. After careful consideration, we feel that it has merit but does not fully meet PLOS ONE’s publication criteria as it currently stands. Therefore, we invite you to submit a revised version of the manuscript that addresses the points raised during the review process.

We look forward to receiving your revised manuscript.

Kind regards,

Mohsin Shafi, Ph.D.

Academic Editor

PLOS ONE

“Funding for the data collection was provided by the Norwegian Nurses Organisation (NSF). No specific funding was received for this publication.”

6. We note that you have indicated that data from this study are available upon request. PLOS only allows data to be available upon request if there are legal or ethical restrictions on sharing data publicly. For more information on unacceptable data access restrictions, please see http://journals.plos.org/plosone/s/data-availability#loc-unacceptable-data-access-restrictions.

Reviewers' comments:

Reviewer's Responses to Questions

**Comments to the Author**

1. Is the manuscript technically sound, and do the data support the conclusions?

Reviewer #1: Yes

Reviewer #2: Yes

2. Has the statistical analysis been performed appropriately and rigorously? 

Reviewer #1: Yes

Reviewer #2: Yes

3. Have the authors made all data underlying the findings in their manuscript fully available?

Reviewer #1: No

Reviewer #2: Yes

4. Is the manuscript presented in an intelligible fashion and written in standard English?

Reviewer #1: Yes

Reviewer #2: Yes

5. Review Comments to the Author

Reviewer #1: Review of PONE-D-22-27893 – The Norwegian version of the Copenhagen Psychosocial Questionnaire (COPSOQ III): Initial validation study using a national sample of licensed nurses

Main justification for the recommended decision

The authors report the results of a validation study of the Copenhagen Psychosocial Questionnaire (COPSOQ III) translated into Norwegian. I enjoyed reading the manuscript, and I think the authors provide a well-informed justification for the focus on the psychosocial work environment, for why there is a need for more translated versions of the COPSOQ, and for the methods they used to assess the translated measurement instrument. The translation and the analyses appear very sound. I commend the authors for their contribution to the literature by improving the chances for building knowledge about the psychosocial work environment across countries. I also think it is good that the authors not only used item response theory models but also factor analyses for triangulation and increased comprehension.

I recommend that the manuscript goes through a minor revision prior to resubmission. My main concern is the content of the introduction, and the general structure of the manuscript. I address each of these in turn, and I also include a list of more specific comments at the end of the introduction.

Revising the introduction

In the introduction, there is no clear, conceptual definition of a psychosocial work environment. Nor is there any detail in the theories about the psychosocial work environment that the authors mention. Finally, there is no description of other scales used for assessing the psychosocial work environment, or for why the COPSOQ is superior.

I think the authors should address the psychosocial work environment as a concept in much more detail in their introduction. What is a psychosocial work environment, i.e., what conceptual dimensions do we assume it consists of? Which theories are used to explain the effects of a psychosocial work environment? What are the psychological explanations of why a better psychosocial work environment reduces sickness absenteeism and increases motivation for work?

I realize that this is not a theoretical article on the psychosocial work environment, but I still believe that some conceptualization and contextualization is necessary in order to give a better description of the context, and in order to situate the COPSOQ instrument relative to our understanding of the psychosocial work environment and relative to alternative scales.

I refer the authors to the following papers as a starting point. I also note that the psychosocial work environment specifically is not my main area of expertise, and that the authors may know of better papers to use as sources and as inspiration for revising their introduction section.

• Nieuwenhuijsen, K., Bruinvels, D., & Frings-Dresen, M. (2010). Psychosocial work environment and stress-related disorders, a systematic review. Occupational Medicine, 60(4), 277-286. https://doi.org/10.1093/occmed/kqq081

• Solovieva, S., Lallukka, T., Virtanen, M., & Viikari-Juntura, E. (2013). Psychosocial factors at work, long work hours, and obesity: a systematic review. Scandinavian Journal of Work, Environment & Health, 241-258. https://www.jstor.org/stable/23558350

• Vazquez, A. C. S., Pianezolla, M., & Hutz, C. S. (2018). Assessment of psychosocial factors at work: A systematic review. Estudos de Psicologia (Campinas), 35, 5-13. https://doi.org/10.1590/1982-02752018000100002

Revising the remainder of the manuscript

Regarding the remainder of the manuscript, I believe that the following sections should be majorly revised or moved to other parts of the manuscript:

• Motive for the translation (lines 333-347) section in the discussion. Parts of this section can either be moved to the introduction or merged with the section on the empirical results. The main summary of the motive for the translation can be summarized much more succinctly.

• The Translation (lines 348-356) section in the discussion belongs in the description of the method. I would recommend moving this section to the Translation section of the Method section (lines 128-141).

• Other COPSOQ studies confirming measurement equivalence (lines 371-398) section in the discussion. This section has a detail level that I believe is unsuitable for the Discussion section, and it is quite unstructured. The authors should work to abstract the most important aspects of the other studies, and more clearly compare these studies, their contexts and their methods of analyses to their own.

• Strengths and limitations (lines 405-409) section in the discussion. This is very brief, and not as informative as it could be. I would recommend that the authors go more into detail about the strengths and limitations of their translation process and their statistical analyses, as these are the most important aspects of their study that touch upon the contribution of the paper (a translated version of the COPSOQ with an initial validation study).

• Further research (lines 410-416) section in the discussion. This section should be expanded following the addition of a new section to the introduction, as per the suggestions in this review. Additionally, it would be helpful with some more specific suggestions for future research also.

More specific comments

• In the abstract

o Considering that the authors focus only on one of three parts of the COPSOQ in their validation study, I think this should be reflected in the abstract, specifically by adding a line about this in the Methods sections of the abstract (lines 31-38).

o The Conclusion section in the abstract should be elaborated on.

• Data availability

o The authors answer that the “Depersonalized data on the 86 items can be provided on request”. What is the justification for not uploading the data to a public repository? This should be included in the description.

• Introduction

o Lines 111-112: The phrasing of this sentence makes the meaning unclear. It may be worth rephrasing it to increase clarity.

o Lines 124-126: These theories are only mentioned by name and not elaborated on. This comment connects to the more general comment concerning a revision of the introduction, and should be seen in relation to that.

• Method

o Lines 148-149: What is the justification for using this cutoff?

o Line 194: Typographical error. This should be “for dimensions that”.

o Line 207: Typographical error. Because the sentence refers to the items, and not the set, it should be “that together provide”.

o Lines 209-212: Wordy sentence, please consider splitting the sentence into two and simplifying these for increased clarity.

o Line 222: Which version of the jamovi software was used?

• Kaiser-Meyer-Olkin measure (Lines 224-230).

o When describing this measure, the authors do not discuss the implication of their sample size for the suitability of the KMO values. Additionally, the reference that the authors refer to for suitability of different values does not support what they describe. This reference, which itself refers to a different reference that cannot be traced, for example refers to KMO values from 0.50 to 0.59 as “miserable”, not as “indicating the adequacy of the sample”. I would recommend that the authors refer to a more transparent source that also considers the role of sample size in determining suitability of KMO values, such as this one: http://pubs.sciepub.com/ajams/9/1/2

• Method (continued)

o Line 232: The Norwegian word “og” should be replaced with the English “and”.

• Results

o Line 243: Typographical error. Should be “Translation”

o Lines 268-270: The authors should go into more detail about how they deal with missing data.

• Discussion

o Lines 349-350: These claims are not substantiated by references and should be.

o Lines 354-356: These descriptions are fairly vague. It would help if the authors exemplified dialect words that were discussed or which specific choices they made in the translation process to keep an informal oral style.

o Lines 408-409: This sentence is unfinished. There is one word, or several words, missing between “COPSOQ III” and “for the entire working population”

o Lines 418-420: This sentence should be divided in two for increased comprehensibility.

Conclusion

Overall, I think this is a very promising manuscript, and I look forward to reading the revised version.

Reviewer #2: The paper is very well written and covers an important area.

I have some (more genral) comments.

1. And some points the authors talk about "the COPSOQ". They should make cleare which of all possible items (CORE, middle, long) were used. All of them?

2. Some of the items were identified as "candidate for removal". Are they from CORE, middle, or long? This information sould be given in the text and could be added in table S1.

3. The study contains only one professional group. This is stated correctly at different points. However it should be made clearer that a lot of statistics (distribution of answers, i.e. dimension ED for a sample of nurses) and some findings maybe due to this fact. Expecially a decision on removal of items can never be based on such a sample (correct in line 294).

4. Table 1: I would like to see some comments on bottom and ceiling effects.

5. Line 331: I would be interested in sme more details what is congruent and what not compared to the English version but also to other validation studies.

6. Line 371ff (up to 398): Here a (short!) comparison to the main stucture findings (factors, dimensions) of other validation studies would be very fruitful (France, Hungary, Germany, Chili, Turkey).

7. Explanation of IRT is lengthy (expecially the formulae from line 180-195).

8. Table S2 and exp. S3. For a better orientation I would suggest to omit small values (in S3 i.e. < 0.3) or (better again) to mark higher values (i.e. > 0.4) in bold.

6. PLOS authors have the option to publish the peer review history of their article (what does this mean?). If published, this will include your full peer review and any attached files.

Reviewer #1: **Yes: **Simen Bø

Reviewer #2: **Yes: **Matthias Nuebling

---

## [Author Response · Author response to Decision Letter 0]

28 Apr 2023

Pleases see uploaded document with our response to reviewers.

---

## [Decision Letter · Decision Letter 1]

26 Jul 2023

The Norwegian version of the Copenhagen Psychosocial Questionnaire (COPSOQ III): Initial validation study using a national sample of licensed nurses

PONE-D-22-27893R1

Dear Dr. Ose,

We’re pleased to inform you that your manuscript has been judged scientifically suitable for publication and will be formally accepted for publication once it meets all outstanding technical requirements.

Kind regards,

Miquel Vall-llosera Camps

Senior Editor

PLOS ONE

Reviewers' comments:

Reviewer's Responses to Questions

**Comments to the Author**

1. If the authors have adequately addressed your comments raised in a previous round of review and you feel that this manuscript is now acceptable for publication, you may indicate that here to bypass the “Comments to the Author” section, enter your conflict of interest statement in the “Confidential to Editor” section, and submit your "Accept" recommendation.

Reviewer #1: All comments have been addressed

Reviewer #2: All comments have been addressed

2. Is the manuscript technically sound, and do the data support the conclusions?

Reviewer #1: Yes

Reviewer #2: Yes

3. Has the statistical analysis been performed appropriately and rigorously? 

Reviewer #1: Yes

Reviewer #2: Yes

4. Have the authors made all data underlying the findings in their manuscript fully available?

Reviewer #1: Yes

Reviewer #2: Yes

5. Is the manuscript presented in an intelligible fashion and written in standard English?

Reviewer #1: Yes

Reviewer #2: Yes

6. Review Comments to the Author

Reviewer #1: (No Response)

Reviewer #2: All my suggestions of the first review round have been met and explained. The paper is now ready for publication. Good luck!

7. PLOS authors have the option to publish the peer review history of their article (what does this mean?). If published, this will include your full peer review and any attached files.

Reviewer #1: **Yes: **Simen Bø

Reviewer #2: **Yes: **Matthias Nübling

---

## [Editor Report · Acceptance letter]

16 Aug 2023

PONE-D-22-27893R1 

The Norwegian version of the Copenhagen Psychosocial Questionnaire (COPSOQ III): Initial validation study using a national sample of registered nurses 

Dear Dr. Ose:

I'm pleased to inform you that your manuscript has been deemed suitable for publication in PLOS ONE. Congratulations! Your manuscript is now with our production department. 

Kind regards, 

on behalf of

Dr. Miquel Vall-llosera Camps 

Staff Editor

PLOS ONE